# Agronomic Performance, Capsaicinoids, Polyphenols and Antioxidant Capacity in Genotypes of Habanero Pepper Grown in the Southeast of Coahuila, Mexico

Neymar Camposeco-Montejo [1], Antonio Flores-Naveda [1], Norma Ruiz-Torres [1], Perpetuo Álvarez-Vázquez [2], Guillermo Niño-Medina [3], Xochitl Ruelas-Chacón [4], María Alejandra Torres-Tapia [1], Pablo Rodríguez-Salinas [5], Victor Villanueva-Coronado [1] and Josué I. García-López [1,*]

[1] Centro de Capacitación y Desarrollo en Tecnología de Semillas, Departamento de Fitomejoramiento, Universidad Autónoma Agraria Antonio Narro, Saltillo 25315, Mexico; neym_33k@hotmail.com (N.C.-M.); naveda26@hotmail.com (A.F.-N.); n_nruiz@hotmail.com (N.R.-T.); atorres_tapia@hotmail.com (M.A.T.-T.); victor.villanueva6913@gmail.com (V.V.-C.)

[2] Departamento de Recursos Naturales, Universidad Autónoma Agraria Antonio Narro, Calzada Antonio Narro 1923, Saltillo 25315, Mexico; perpetuo.alvarezv@uaaan.edu.mx

[3] Facultad de Agronomía, Universidad Autónoma de Nuevo León, Francisco Villa S/N, Col. Ex-Hacienda el Canada, General Escobedo 66050, Mexico; guillermo.ninomd@uanl.edu.mx

[4] Departamento de Ciencia y Tecnología de Alimentos, Universidad Autónoma Agraria Antonio Narro, Calzada Antonio Narro 1923, Saltillo 25315, Mexico; xruelas@uaaan.edu.mx

[5] Departamento en Ciencias Biológicas, Facultad de Ciencias Naturales, Universidad Autónoma de Querétaro, Avenida de las Ciencias S/N, Juriquilla 76230, Mexico; palanvf@hotmail.com

\* Correspondence: g.lopezj90@gmail.com or josue.garcia@uaaan.edu.mx

**Abstract:** The genetic improvement program of the Seed Technology Training and Development Center works on the agronomic characterization and the content of bioactive compounds in eight genotypes of habanero pepper. The objective is to select genotypes with good agronomic performance that allow the generation of inbred lines to obtain hybrids. In this study, the agronomic performance and the content of bioactive compounds (capsaicinoids, polyphenols, and antioxidant capacity) were evaluated in eight genotypes of habanero pepper grown in the southeast of Coahuila, Mexico, identified as HNC-1, HNC-2, HNC-3, HNC-4, HNC-5, HNC-6, HNC-7, and HCC-8. The plants were grown in a greenhouse for 127 days, under a completely randomized design with four replications each. The results revealed that the yield (g·plant$^{-1}$) and number of fruits per plant did not show significant differences between genotypes. However, for the fruit length, the genotypes HCC-8, HNC-7, HNC-6, and HNC-5 stood out with over 40 mm, while in equatorial diameter of the fruit, HCC-8, HNC-4, and HNC-2 stood out with 26.45, 26.46, and 25.12 mm, respectively. The results of the chemical analyses allowed us to identify that HNC-5 and HNC-6 had the highest capsaicin concentration (931.38 and 959.77 mg·kg$^{-1}$), dihydrocapsaicin (434.95 and 445.89 mg·kg$^{-1}$), Scoville Heat Units greater than 210,000, total phenols (67.54 and 71.15 mg/100 g) and total flavonoids (34.21 and 38.29 mg/100 g), respectively. The HNC-1 and HNC-6 genotypes had the highest carotenoids concentration with 103.96 and 105.07 mg/100 g, and HCC-8 registered the highest anthocyanin content with 22.08 mg C3GE/100 g. The antioxidant capacities showed significant differences ($p \leq 0.05$) between genotypes, with a range of 43.22 to 110.39 μmol TE/100 g, 72.37 to 186.56 μmol TE/100 g, and 191.41 to 244.98 μmol TE/100 g for the tests of DPPH (2,2-diphenyl-1-picrylhydrazyl), ABTS (2,2′azino-bis(3-ethylbenzothiazoline-6-sulfonic acid)), and FRAP (ferric reducing antioxidant power). The results of this research will be used to select habanero pepper genotypes that can be used in genetic improvement programs to increase the productive potential and the content of bioactive compounds in the fruits to expand their applications in the food industry.

**Keywords:** *Capsicum chinense* Jacq.; fruit yield; pungency; bioactive compounds; antioxidant capacity

## 1. Introduction

Mexico is one of the major centers of origin and domestication of the *Capsicum* genus [1]. To date, 32 taxa have been described, of which 5 have been domesticated and cultivated: *C. annuum* L., *C. frutescens* L., *C. baccatum* L., *C. pubescens*, and *C. chinense* Jacq. [2,3]. The habanero pepper (*C. chinense* Jacq.) is a species within this genus that has the highest demand in the market and is one of the most pungent chili peppers in the world [4].

The ingredients derived from the habanero pepper, in addition to being used as a condiment, providing pungency, color and characteristic flavor, could be used to conserve and prolong the useful life of industrial products, as well as additives or technological ingredients with antioxidant and antimicrobial activities [5]. To treat inflammatory conditions, new products have been developed since habanero peppers contain high levels of capsaicinoids, carotenoids, phenolic compounds, vitamin C and A, and minerals, such as iron and calcium, which promote human health [5,6].

Several studies have documented that the main phenolic compounds found in peppers are vanillic, caffeic, ferulic, *p*-coumaric, and *p*-hydroxybenzoic acids [7]. Ferulic acid has antiradical properties and vanillic acid is used primarily as a flavor enhancer [8]. Furthermore, it is estimated that capsaicin and dihydrocapsaicin represent 80% of capsaicinoids and their amounts are largely determined by the level of pungency [9].

Capsaicinoids demonstrate anti-inflammatory activity, in addition to promoting energy consumption and suppressing fat accumulation [10]. The antioxidant activities of capsaicinoids and their lack of acrimony make them attractive for potential applications in food and pharmacology [11]. It should be noted that there is a positive correlation between antioxidant activity and the concentration of phenolic compounds in habanero pepper fruits [4]. This suggests that programs aimed at the genetic improvement of horticultural crops should consider the development of peppers with a high concentration of bioactive compounds that can be used in the food and pharmacological industry.

In Mexico, there is a regional trend to produce and consume habanero peppers. The main production areas are in the states of Yucatán, Campeche, and Quintana Roo [12]. However, despite the economic, social and cultural importance of the crop, few efforts have been made to improve varieties of habanero peppers to increase the yield and the content of bioactive compounds in the fruits (polyphenols and carotenoids), besides cultivating this crop in other regions of Mexico and in other parts of the world [3].

Currently, there is great interest in developing the commercial exploitation of this crop in northeastern Mexico since, according to the Mexican Food and Agriculture Information Service (SIAP), of the 1134-ha planted in 2020, 8% were grown in the states of Coahuila, Chihuahua, Nuevo León, Tamaulipas, and Aguascalientes, with an average yield of 18 tons per ha [13].

The demand of the habanero pepper market is increasing [14]. However, the problems that cultivation faces to satisfy this demand continues to be the low production technology since, of the total area sown, more than 800 hectares are open field [13]. Another problem is the use of landraces degraded in their varietal purity and seed quality, the main causes of poor crop yields, in addition to pests, diseases, inadequate crop nutrition and water deficit due to inefficient irrigation systems [15,16].

Therefore, it is necessary to create new techniques that increase the yields of habanero peppers per unit of cultivated area, meeting the expectations of producers and consumers. One of these techniques is plant genetic improvement, a method that has been widely used since pre-Hispanic times to improve food production for humanity [17]. In *Capsicum*, as in other horticultural species, obtaining crop varieties has generally been carried out through the evaluation and selection of promising genotypes within the populations and the existing genetic diversity, or by following the schemes of classical genetic improvement and hybridization [17,18].

In this regard, is important to use the existing genetic diversity to produce improved varieties of habanero pepper that can adapt to new local and regional agroclimatic con-

ditions to enhance high productivity. Therefore, the objective of this work was to evaluate the agronomic performance and the content of bioactive compounds (capsaicinoids, polyphenols and antioxidant capacity) in eight genotypes of habanero pepper grown in the southeast of Coahuila, Mexico, to select them and include them in improvement programs.

## 2. Materials and Methods

### 2.1. Experimental Site Location

The experiment was carried out in a greenhouse of $9 \times 20$ m, with an average daily temperature of 28.7 °C and a relative humidity of 68%, belonging to the Plant Breeding Department of the Universidad Autónoma Agraria Antonio Narro (UAAAN), in Saltillo, Coahuila, at 25°21′24″ LN and 101°02′05″ LO, and 1762 m above sea level.

### 2.2. Genetic Material and Seedling Production

The seeds of habanero pepper (HNC-1, HNC-2, HNC-3, HNC-4, HNC-5, HNC-6, HNC-7, and HCC-8), belong to the germplasm that protects the Seed Technology Training and Development Center (CCDTS) of the UAAAN (Table 1). For sowing, the seeds were treated with a 250-ppm gibberellic acid solution for 24 h as a germination inducer, later; they were sown in polystyrene trays of 200 cavities; the germination substrate was peat moss and perlite in a 70:30 proportion.

**Table 1.** Characteristics of the habanero pepper genotypes selected for the study.

| Genotype | Color | Origin | Type | Selection Cycle in Coahuila | Picture |
|---|---|---|---|---|---|
| HNC-1 | Orange | Yucatán | Creole | 2 |  |
| HNC-2 | Orange | Jaguar | Selection | 3 |  |
| HNC-3 | Orange | Jaguar | Selection | 3 |  |
| HNC-4 | Orange | Jaguar | Selection | 3 |  |
| HNC-5 | Orange | Yucatán | Creole | 2 |  |
| HNC-6 | Orange | Yucatán | Creole | 2 |  |

**Table 1.** *Cont.*

| Genotype | Color | Origin | Type | Selection Cycle in Coahuila | Picture |
|---|---|---|---|---|---|
| HNC-7 | Orange | Jaguar * | Selection | 3 |  |
| HCC-8 | Chocolate | Coahuila | Creole | 7 |  |

* Jaguar, an open-pollinated variety with registration number CHL-008-101109 and Breeder's Title No. 0664 from the National Institute for Agricultural and Livestock Forestry Research by Ramírez Meraz et al. [19].

*2.3. Crop Management*

The sowing of the seeds was carried out on 20 April 2020. The transplant was carried out on 13 June 2020, and 54 days after sowing the seedlings were grown in 10 L pots using a mixture of peat moss and perlite in a 70:30 ratio as substrate. Water and nutrients were supplied by irrigation at five cm from the base of the stem. The nutrient solution used is shown in Table 2 (50% at five days after transplantation, 75% at the beginning of flowering and 100% at fruiting and filling), and the drainage was 10%.

**Table 2.** Nutritive solution (NS) and percentages used in each of the phenological stages of the cultivation of habanero pepper under greenhouse conditions.

| NS% | $NO_3^-$ | $H_2PO_4^-$ | $SO_4^{2-}$ | $Cl^-$ | $HCO_3^-$ and $CO_3^{2-}$ | $NH_4^+$ | $K^+$ | $Mg^{2+}$ | $Ca^{2+}$ | $Na^+$ |
|---|---|---|---|---|---|---|---|---|---|---|
| Fructification-100 | 12 | 2.0 | 7.0 | 3.2 | 1.0 | 2.0 | 7.0 | 4.2 | 9.0 | 3.1 |
| Flowering-75 | 9.0 | 1.5 | 5.3 | 3.2 | 1.0 | 1.5 | 5.3 | 3.2 | 6.8 | 3.1 |
| Trasplant-50 | 6.0 | 1.0 | 3.5 | 3.2 | 1.0 | 1.0 | 3.5 | 2.1 | 4.5 | 3.1 |

Milliequivalents $L^{-1}$.

*2.4. Fruit Yield Components*

The first harvest was carried out on 10 September (85 days after transplantation), the second on 1 October and the third on 22 October 2020. The total fruit weight was determined by weighing all of the fruits of the plot of each harvest on a Sartorius precision digital scale (TS 1352Q37, Göttingen, Germany). The plot yield was divided by the number of plants to obtain the yield per plant (g·plant$^{-1}$), and later the number of fruits was counted. The average fruit weight (g) was calculated dividing the total fruit weight between the total number of fruits, fruit length and equatorial diameter (mm) by randomly taking eight fruits per plot in each harvest, using a Truper digital vernier (CALDI-6MP, Atlacomulco, Mexico).

*2.5. Agronomic Parameters*

The first harvest was carried out on 10 September (85 days after transplantation), the second on 1 October and the third on 22 October 2020. The total fruit weight was determined by weighing all of the fruits of the plot of each harvest on a Sartorius precision digital scale (TS 1352Q37, Göttingen, Germany). Plant height (cm) and root length (cm) were determined with a Truper tape measure (PRO-5MEC, Atlacomulco, Mexico). The plot yield was divided by the number of plants to obtain the yield per plant (g·plant$^{-1}$), later the number of fruits was counted. The average fruit weight (g) was calculated dividing the total fruit weight between the total number of fruits, fruit length and equatorial diameter

(mm) by randomly taking eight fruits per plot in each harvest, using a Truper digital vernier (CALDI-6MP, Atlacomulco, Mexico).

### 2.6. Physicochemical Analysis of Habanero Peppers

From the last harvest, four replicates of 50 fruits were selected for each genotype [20]. Before the functional analyzes, the fruits were washed with a 3% NaClO solution [21]. Of the total of the selected fruits, 25 fruits were used to determine the chromatic properties of the fruit, and 25 fruits for functional analyzes (capsaicinoids, polyphenols, and antioxidant capacity) [22].

### 2.7. Fruit Color Tests

The color characteristics were determined on the exocarp of the fruit, with a Konica Minolta colorimeter (CR-10, Tokyo, Japan), according to García-López et al. [4]. The color parameters were evaluated using the CIELCH coordinates ($L$ *, $C$ * and $h$), established per the Commission Internationale De L'ecleirage (CIE) [23]. $L$ * defines the luminosity (0 black, 100 white), $C$ * (chrome; $h$ saturation level) and $h$ * (hue angle: $0°$ = red, $90°$ = yellow, $180°$ = green, $270°$ = blue). The color graphs were made with the online software ColorHexa [24], using the values obtained in $L$ *, $C$ * and $h$ *.

### 2.8. Functional Analysis

To quantify the bioactive compounds (polyphenols and antioxidant capacity), the habanero pepper fruits were cut into slices and the seeds were removed. The samples were stored at $-80\,°C$ until the determinations were carried out.

#### 2.8.1. Capsaicinoid Extraction

The extraction and quantification of capsaicinoids (capsaicin and dihydrocapsaicin) was carried out based on the work of Ryu et al. [25].

#### 2.8.2. Quantification of Capsaicin and Dihydrocapsaicin by High Performance Liquid Chromatography (HPLC)

The capsaicinoid extracts were injected (10 µL) into the Agilent Technologies 1260 Infinity HPLC, equipped with a quaternary pump (Agilent 1260 G1311B), autosampler and diode array detector (DAD, Agilent 1260 G4212B). The separation column was ZORBAX Eclipse Plus C-18 (100 mm × 3 mm id, 5 µm). For the separation of compounds, an isocratic mobile phase was used (pure acetonitrile and 1% acetic acid in water, in a ratio of 40:60 $v/v$), with a flow rate of 1 mL min$^{-1}$ at 25 °C for 20 min.

Capsaicinoid concentration was determined based on calibration curves at concentrations of 0, 80, 160, 240, 320 mg·kg$^{-1}$, using reference standards (capsaicin 8-Methyl-N-vanillyl-trans-6-nonenamide and 8-Methyl-N-vanillylnonanamide) from Sigma Aldrich, St. Louis, MO, USA [26].

#### 2.8.3. Determination of Scoville Units (SHU)

Scoville heat units (SHU) for each genotype were calculated based on what was established per Todd et al. [27].

#### 2.8.4. Extraction and Quantification of Phenolic Compounds

The extraction and quantification of total phenols, total flavonoids, total anthocyanins, and carotenoids were carried out in a Thermo Spectronic BioMate3 spectrophotometer (Rochester, NY, USA), following the methods described by García-López et al. [4] and Abdel-Aal and Hucl [28].

### 2.9. Antioxidant Capacity

The antioxidant capacity tests for DPPH, ABTS, and FRAP were carried out according to López-Contreras et al. [29]. The results were reported in microchromols of

Trolox (6-hydroxy-2,5,7,8-tetramethylchroman-2-carboxylic acid) equivalents per hundred grams of sample (µmol TE/100 g), using as reference the calibration curve of Trolox (0 to 500 µmol/L).

### 2.10. Experimental Design and Statistical Analysis

The experimental arrangement was completely randomized with eight treatments (genotypes evaluated) and four replications. The statistical difference between genotypes was analyzed using an analysis of variance (ANOVA), and the means were compared with the Tukey test ($p \leq 0.05$) using the statistical package SPSS version 21.0 (SPSS Inc., Chicago, IL, USA). Results were reported as mean values of four samples ± standard deviation.

## 3. Results and Discussion

### 3.1. Fruit Yield Components

For yield (g·plant$^{-1}$) and number of fruits per plant, no significant differences were found (Figure 1A,B). However, the performance of the selected genotypes of the Jaguar variety showed a similar behavior, but lower than the genotypes from Yucatán, indicating a possible genetic potential of the HNC-5 and HNC-6 genotypes under the evaluated conditions. It should be noted that the mean yields of the evaluated genotypes coincide with those reported by Loatournerie et al. [30] in genotypes evaluated in Conkal, Yucatán in 2009. The number of fruits per plant (Figure 1B) in the HNC-5 and HNC-6 genotypes also stood out, as well as the HNC-3 and HNC-2 genotypes, which are selections of the Jaguar variety, a character that could be improved through selection and hybridization [31].

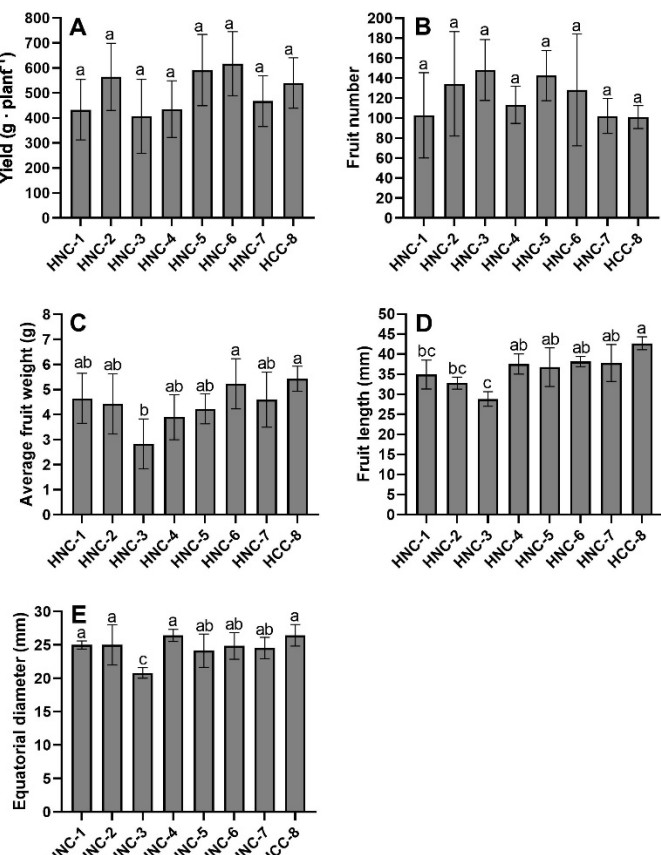

**Figure 1.** Yield (**A**), number of fruits (**B**), average fruit weight (**C**), fruit length (**D**), and equatorial diameter (**E**) of eight genotypes of habanero pepper grown in the southeast of Coahuila. Values are the average of four replications. Means (*n* = 4). Bars represent the standard deviation of the mean. Different letters in each bar mean that the genotypes were statistically different (Tukey, *p* ≤ 0.05).

For the average fruit weight, a similar statistical behavior was observed between genotypes (Figure 1C). The lowest weight was obtained in HNC-3, while the highest values were observed in the HCC-8 and HNC-6 genotypes (with over 5 g), the first genotype is distinguished by being chocolate-colored, and the second for having orange fruits directly from Yucatán (Table 1), place of dispersal and greater genetic diversity of the species [18].

In fruit length, the HCC-8 genotype stands out with over 40 mm in length, however, in the same statistical group are the genotypes HNC-7, HNC-6, HNC-5, HNC-4, and HNC-1 (Figure 1D). The equatorial diameter of the fruit was similar between genotypes, except for HNC-3, which had the smallest diameter (20.8 mm) (Figure 1E). The results described are similar to those reported by Tucuch et al. [32] and Peña [33], who found average values of 25.4 mm for the equatorial diameter of the fruit in habanero pepper genotypes. However, Ramírez Meraz et al. [19] pointed out that the Jaguar variety produces fruits with a length ranging from 3.8 to 5.5 cm, and diameters between 2.5 and 3 cm, while the average weight of the fruit was 6.5–10 g, values that are higher than those obtained in this work.

In that context, it has been documented that habanero peppers produced under greenhouse conditions are generally smaller than those produced in the open field, which is attributed to a lower incidence of solar radiation inside greenhouses [34]. For their part, Tapia et al. [35], point out that the fruit length of the chocolate-colored habanero pepper is 2.92 cm, and its diameter is 2.44 cm when it is produced under greenhouse conditions, results lower than those obtained in this research for the genotype HCC-8 (chocolate habanero), which can be attributed to the agroclimatic conditions of the evaluated area.

### 3.2. Agronomic Parameters

The results of means comparison of the agronomic variables are shown in Figure 2. For plant height, a similar behavior was observed between genotypes, except for HNC-1 and HNC-5, which were the lowest (Figure 2A). These results are superior to those reported by López Arcos et al. [36], but lower than those reported by López-Gómez et al. [37]. The differences observed in some of the evaluated characteristics are attributed to the origin and genetics of the materials, characteristics that could be improved through selection and inbreeding [31], although the tested environment also influences in the phenotypic characteristics observed [30]. The variability in agronomic behavior between genotypes could guide the selection of materials with the best phenotypic characteristics and include them in genetic improvement programs, creating varieties adapted to new agroclimatic conditions or specific environments [3,38], since the previously described *C. chinenese* characters are generally highly heritable [39,40]. In addition, genetic diversity can also provide the guidelines to exploit heterosis to generate superior hybrids [41].

In the variable of basal stem thickness, the genotype that showed the best performance was HNC-6 with 14 mm, followed by HNC-2, HNC-3, and HNC-7 (Figure 2B), a greater stem thickness is generally associated with a greater transport of nutrients and sap produced by the conducting vessels, which translates into a greater foliar area, stem length and accumulation of fresh weight [42]. The results found in this variable are more than 30% higher than those reported by López Arcos et al. [36], albeit 30% lower than those reported by López-Gómez et al. [37].

The length of the internodes was statistically similar between the genotypes HNC-3, HNC-7, HNC-8, and HNC-1, while the shortest internodes were found in HNC-4, HNC-5, and HNC-6 (Figure 2C). It is important to note that HNC-5 and HNC-6 come from Yucatán, which could be included in the genetic improvement programs to obtain more compact plants with a concentrated production or otherwise use HNC-3, HNC-7, HNC-8 and HNC-1 for open plants and a continuous production.

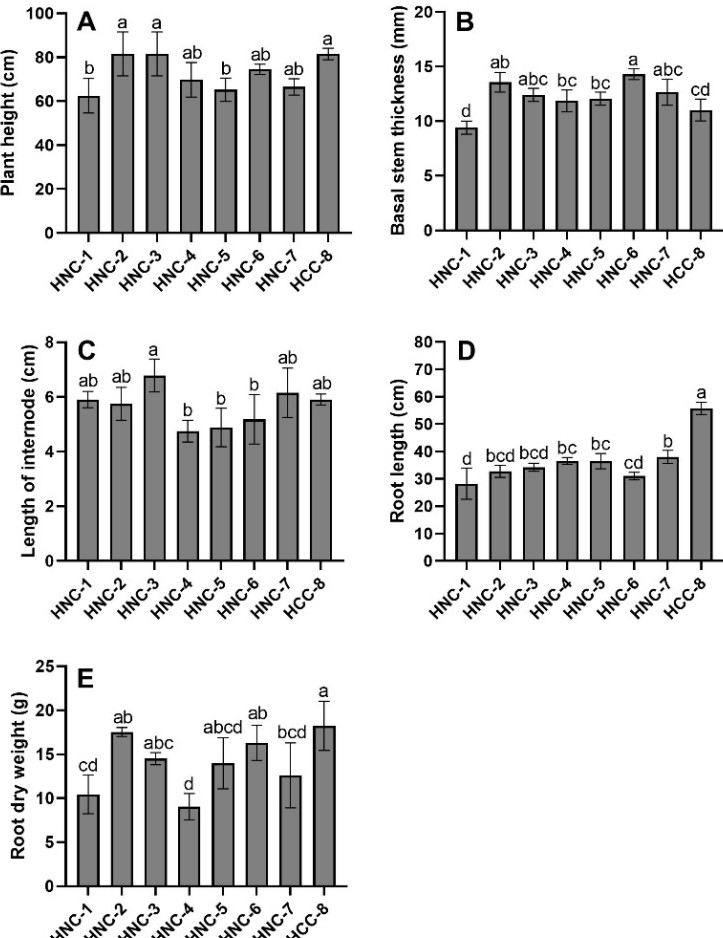

**Figure 2.** Plant height (**A**), basal stem thickness (**B**), internode length (**C**), root length (**D**), and root dry weight (**E**) of eight genotypes of habanero pepper grown in the southeast of Coahuila. Values are the average of four replications. Means (*n* = 4). Bars represent the standard deviation of the mean. Different letters in each bar mean that the genotypes were statistically different (Tukey, *p* ≤ 0.05).

The root length was higher in the HCC-8 genotype since it exceeded the rest of the genotypes by more than 46% (Figure 2D). In the accumulation of root dry weight, a statistically similar behavior was observed in the genotypes HCC-8, HNC-2, and HNC-6, with 18.23, 17.53 and 16.30 g, respectively, while those with less accumulation of dry matter in root were HNC-4 and HNC-1 with 9.03 and 10.14 g, respectively (Figure 2E). In this regard, López-Gómez et al. [37] found a root dry weight of 60 g when grown in tezontle, however, under greenhouse conditions and using artificial substrates, growth, and productivity of plants was achieved with a reduced root system [43].

### 3.3. Fruit Color

The color characteristics of the habanero pepper fruits showed significant differences (*p* ≤ 0.05) between genotypes (Table 3). Regarding the luminosity (*L* \*), the values ranged between 26.88 and 51.56, where the genotypes HNC-5 and HNC-7 (with fruits of moderate orange color), showed the highest values of luminosity (51.06 and 51.56, respectively), while the genotypes HNC-6 (dark orange-brown fruits) and HNC-8 (dark grayish-red fruits) had the lowest luminosity values with 40.28 and 26.88, respectively (Table 3).

These results were similar to those reported by Vera-Guzmán et al. [44], who reported *L* \* values in a range from 21.8 to 52.8 in wild genotypes of *Capsicum annuum* L cultivated in Oaxaca, Mexico. Regarding the color characteristics of *C. chinense*, these present variants that range from orange due to the content of zeaxanthin, *β*-cryptoxanthin and *β*-carotene, to

an intense red induced by capsanthins and capsorubin [1,45]. However, the fruit color can be affected by different factors, including light availability (which regulates morphological characteristics and acts as an energy source for metabolism and photosynthetic processes), temperature, climatic conditions, genotype, minerals supply, growth conditions, and maturity stage (during the fruit ripening stage, biochemical, physiological and structural changes occur, which determine the characteristics of the final fruit) [12,46].

For the case of *C* *, the results showed values ranging from 6.30 to 55.32, located in the gray area of the hue circle, while the lowest value corresponded to HNC-8, the other genotypes were statistically the same (Table 3). It should be noted that within the CIELCH color model, the lower values associated with *C* * indicate a lower color purity, while higher values (close to 50) represent a purer color. Therefore, the lower the saturation of a given color, the greater the grayish hue and discoloration [47].

**Table 3.** Chromatic parameters of the fruits of eight genotypes of habanero pepper grown in the southeast of Coahuila.

| Genotypes | Color Parameters | | | |
|---|---|---|---|---|
| | *L* * | *C* * | *h* * | View |
| HNC-1 | 42.40 ± 0.56 [c] | 55.32 ± 0.99 [a] | 44.86 ± 0.63 [c] | |
| HNC-2 | 47.64 ± 1.35 [b] | 54.66 ± 1.76 [a] | 45.52 ± 1.46 [c] | |
| HNC-3 | 48.38 ± 0.92 [b] | 53.06 ± 1.50 [a] | 45.66 ± 1.69 [bc] | |
| HNC-4 | 48.36 ± 2.21 [b] | 54.46 ± 1.59 [a] | 45.06 ± 2.03 [c] | |
| HNC-5 | 51.06 ± 1.55 [a] | 52.08 ± 2.84 [a] | 51.90 ± 3.10 [a] | |
| HNC-6 | 40.28 ± 0.75 [c] | 54.12 ± 3.20 [a] | 51.15 ± 1.99 [a] | |
| HNC-7 | 51.56 ± 1.19 [a] | 53.14 ± 1.70 [a] | 50.64 ± 1.28 [ab] | |
| HCC-8 | 26.88 ± 0.63 [d] | 6.30 ± 1.81 [b] | 32.04 ± 4.81 [d] | |

*L* *: luminosity; *C* *: saturation level; *h* *: hue angle. Values are the average of four replications. Means ($n$ = 4) ± standard deviation. Different letters within each column mean that the treatments were statistically different (Tukey, $p \leq 0.05$).

The hue angle (*h* *) readings were found in the range from 32.04 to 51.91 (Table 3), according to these values, the color of the analyzed samples is red-yellow, a tone that is associated with a *h* * value of 90° on the hue circle. However, despite the fact that there is not a defined classification nomenclature for the color of the habanero pepper genotypes according to their chromatic values, based on the values obtained in *L* *, *C* * and *h* *, the habanero pepper genotypes are classified into three color groups: (1) Dark orange, which includes HNC-1 and HNC-6; (2) Moderate orange, including HNC-2, HNC-3, HNC-4, HNC-5 and HNC-7; and (3) Dark grayish red, which includes HNC-3.

Muñoz-Ramírez et al. [3] and Fonseca et al. [48], studied the genetic diversity of habanero pepper genotypes and observed that red and orange colors are the predominant colors of ripe fruit in most of the genotypes evaluated. In the fruits of *C. chinense*, it is important to determine the parameters *L* *, *C* * and *h* *, since the pigments responsible for the color (carotenoids and anthocyanins) could be indicative of a higher antioxidant capacity [49,50].

*3.4. Capsaicinoid Content*

The results obtained from the analysis of the capsaicinoid content in the fruits showed significant differences ($p \leq 0.05$) between genotypes (Figure 3). The capsaicinoid concentration ranged between 595.48 and 959.77 mg·kg$^{-1}$ (Figure 3A). The highest content was obtained in HNC-6 (959.77 mg·kg$^{-1}$), followed by HNC-5 (931.38 mg·kg$^{-1}$), while HCC-8 had the lowest concentration (595.48 mg·kg$^{-1}$).

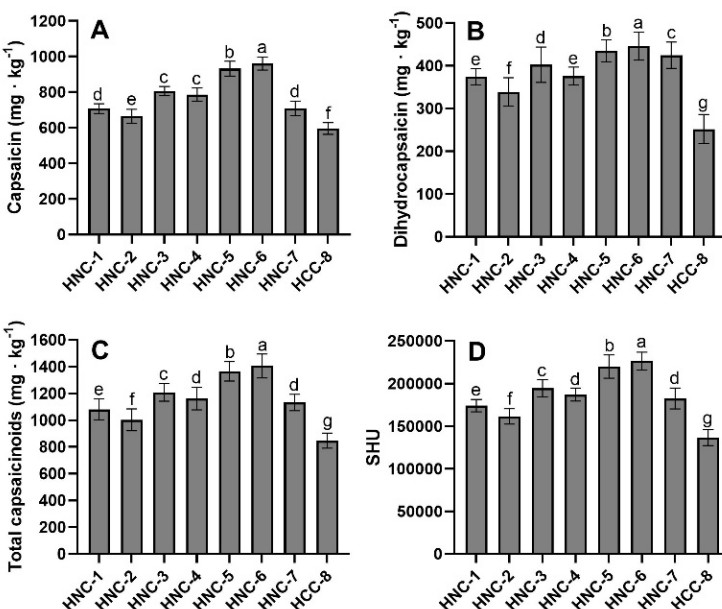

**Figure 3.** Content of capsaicin (**A**), dihydrocapsaicin (**B**), total capsaicinoids, (**C**) and Scoville Heat Units SHU (**D**) in fruits of eight genotypes of habanero pepper grown in the southeast of Coahuila. Values are the average of four replications. Means (*n* = 4). Bars represent the standard deviation of the mean. Different letters in each bar mean that the genotypes were statistically different (Tukey, $p \leq 0.05$).

The same trend was observed in the dihydrocapsaicin content, the genotypes with the highest concentration were HNC-6 and HNC-5 with 445.89 and 434.95 mg·kg$^{-1}$, respectively (Figure 3B). The quantitative analysis of capsaicinoids by HPLC allowed to identify that the highest content of total capsaicinoids was obtained in the genotypes HNC-6 (1405.67 mg·kg$^{-1}$) and HNC-5 (1366.34 mg·kg$^{-1}$), which resulted in the highest levels of SHU with 226,631 and 219,981, respectively (Figure 3C,D).

The total capsaicinoid content was higher than the reported by García-López et al. [4] and Butcher et al. [51], who reported that total capsaicinoid concentrations in habanero pepper genotypes fluctuated between 757.69 and 952.15 mg·kg$^{-1}$.

Pungency is an exceptional attribute that distinguishes the habanero pepper from other plant species. This characteristic allows it to be used regularly in different cultures, gastronomies, and industrial processes, which has generated a growing demand in both Mexican and international markets [14].

However, the accumulation of total capsaicinoids in the fruits is determined by the genotype and the growth environment [52,53]. Commonly, the SHU scale is used to determine the hotness of peppers, which depends on the concentration of capsaicin and dihydrocapsaicin [26]. SHU scale is classified as (1) not hot (0–700 SHU), (2) slightly hot (700–3000 SHU), (3) moderately hot (3000–25,000 SHU), (4) very hot (25,000–70,000 SHU), and (5) extremely hot (>80,000 SHU) [4]. Based on the results obtained, the habanero pepper fruits of the genotypes under study are classified as very spicy fruits, since most of them exceed the 150,000 SHU.

The determination of the capsaicinoid content in the habanero pepper fruits by HPLC was based on the retention period and on the size of the peak of each capsaicinoid, which was identified by the absorption spectrum and comparison with the retention periods of the commercial standards for each compound [4,52]. The chromatograms obtained showed two main peaks, identified as capsaicin and dihydrocapsaicin, which registered a difference of 3.25 min between the retention periods of capsaicin (6.24 min) and dihydrocapsaicin (9.49 min) (Figure 4).

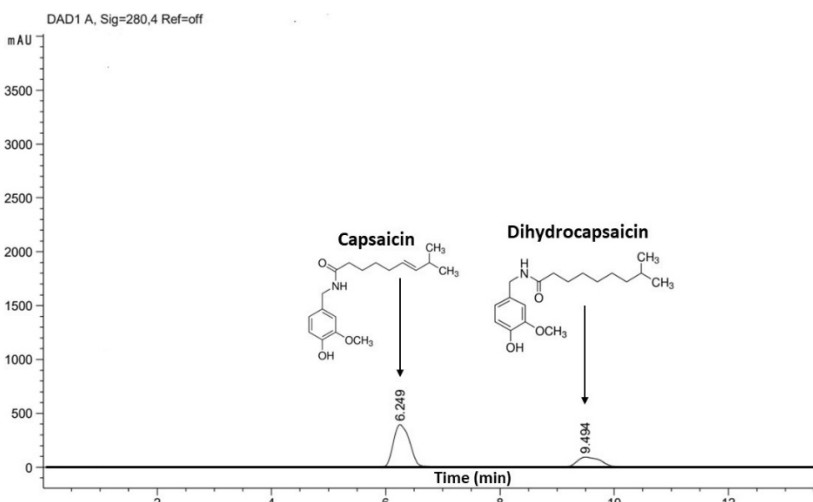

**Figure 4.** Chromatogram of capsaicin and dihydrocapsaicin in habanero pepper fruits grown in the southeast of Coahuila.

### 3.5. Polyphenol and Carotenoid Content

The results showed significant differences ($p \leq 0.05$) between genotypes in the content of total polyphenols (phenols, flavonoids, and anthocyanins) and carotenoids in the fruits (Figure 5). The HNC-5 and HNC-6 genotypes showed the highest levels of total phenols (67.54 and 71.15 mg GAE/100 g, respectively) and total flavonoids (34.21 and 38.29 mg CE/100 g, respectively), while the lowest concentrations were obtained in HNC-2 with 34.87 mg GAE/100 g and 11.40 mg CE/100 g, respectively (Figure 5A,B). The values obtained for most of the samples agree with the results reported by Iqbal et al. [54,55], who evaluated genotypes of hot pepper peppers grown at the Institute of Horticultural Sciences (University of Agriculture, Faisalabad, Pakistan) and reported values in a range of 11.90 to 67.90 mg/100 g.

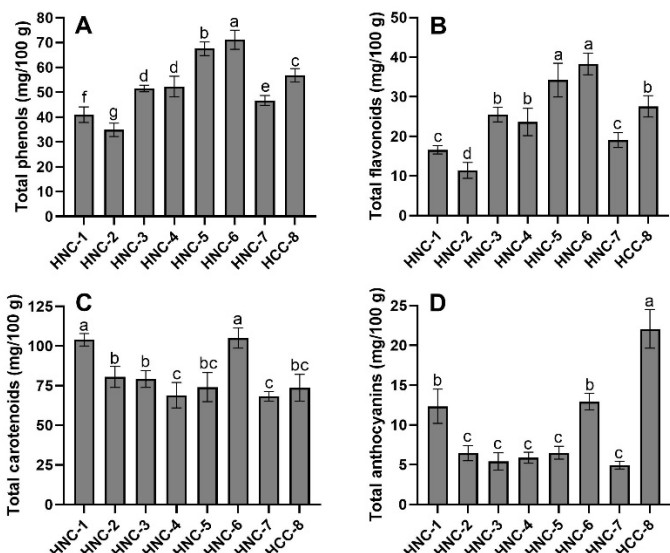

**Figure 5.** Content of total phenols (**A**), total flavonoids (**B**), total carotenoids, (**C**) and total anthocyanins (**D**) in fruits of eight genotypes of habanero pepper grown in the southeast of Coahuila. Values are the average of four replications. Means (*n* = 4). Bars represent the standard deviation of the mean. Different letters in each bar mean that the genotypes were statistically different (Tukey, $p \leq 0.05$).

The content of polyphenols in the fruits varies widely during growth and ripening. Additionally, they determine the pungency of the fruit, bitterness, flavor, and color [6]. In this study, the most intensely colored genotypes (HNC-6 and HCC-8) had the highest content of polyphenols (flavonoids and anthocyanins), which is consistent with previous reports that registered a high content of phenols in *C. chinense* genotypes with orange-brown and deep red fruits [56,57].

The carotenoid content ranged from 68.27 to 105.07 mg/100 g. Among the highest levels of carotenoids, the HNC-1 and HNC-6 genotypes had 103.96 and 105.07 mg/100 g, while the lowest concentrations were recorded in HNC-4 and HNC-7 (69.02 and 68.27 mg/100 g, respectively). The results of this research agree with the average values (34.97 to 118.22 mg/100 g) that have been reported in different genotypes of hot peppers [55,58], with fruit pigmentation ranging from yellow, orange, red and/or dark red.

The red pepper is unique due to its high content and composition of xanthophylls, comprising yellow carotenoids ($\beta$-carotene, violaxanthin, antheraxanthin, zeaxanthin) and the characteristic intense red ketocarotenoids, which are capsanthin, capsorubin, and capsanthin-5,6-epoxide, as well as anthocyanins [58,59]. In yellow/orange pepper peppers, the main carotenoids include lutein, $\beta$-carotene, violaxanthin, antheraxanthin, zeaxanthin, and $\beta$-cryptoxanthin [60,61]. Carotenoids and/or polyphenols are considered the most important antioxidants in habanero pepper, since they act synergistically as free radical scavengers by donating hydrogen atoms to form relatively stable radicals, which prevents oxidative damage [58,62].

The anthocyanin content in pepper fruits was between 4.95 and 22.08 mg C3GE/100 g. The highest anthocyanin content was recorded in the HCC-8 genotype (22.08 mg C3GE/100 g), followed by HNC-6 and HNC-1 (12.91 and 12.35 mg C3GE/100 g), while the lowest concentration was obtained in HNC-7 (4.95 mg C3GE/100 g). The results were similar to the average concentration (3.20 to 27.8 mg/100 g) reported in different pepper genotypes with yellow, orange, and red fruits [59,63]. In some species of hot peppers, the purple color is provided by the accumulation of anthocyanins [64], which are beneficial to human health, due to their antioxidant properties that could potentially prevent harmful diseases such as cancer, cardiovascular problems, and even neurodegenerative diseases [65].

In addition, anthocyanins extracted from seeds and fruits have been used as food additives, mainly in the production of jams, sweets, and violet-colored drinks. These applications in the food industry have awakened a great interest to find fruits with a high concentration of natural food dyes such as anthocyanin as a promising alternative to synthetic food dyes [66]. In this sense, increasing the anthocyanin content in habanero pepper fruits could improve the nutraceutical quality and, therefore, increase the value of the product. On the other hand, the concentration of anthocyanins determines the postharvest quality of some flowers, fruits, and vegetables, so plant breeders have focused their efforts on generating crops rich in anthocyanins for the market [67].

### 3.6. Antioxidant Capacity

The results indicate that the antioxidant capacities of the fruits showed significant differences ($p \leq 0.05$) between genotypes (Figure 6), with a range of 43.22 to 110.39 $\mu$mol TE/100 g, 72.37 to 186.56 $\mu$mol TE/100 g and 191.41 to 244.98 $\mu$mol TE/100 g for the DPPH, ABTS and FRAP assays, respectively (Figure 6). Regardless of the assay, the HNC-6, HNC-5, and HCC-8 genotypes presented the highest levels of antioxidant capacity, while HNC-1, HNC-2 and HCC-7 showed the lowest values. The significant correlations between the total phenol content (r = 0.94, 0.95 and r = 0.88, respectively), total flavonoids (r = 0.93, 0.94 and r = 0.89, respectively) and the antioxidant capacity measured by DPPH, ABTS and FRAP (Figures 5 and 6) suggests that the higher values of phenolic compounds in HNC-6, HNC-5, and HCC-8 are related to a higher antioxidant capacity.

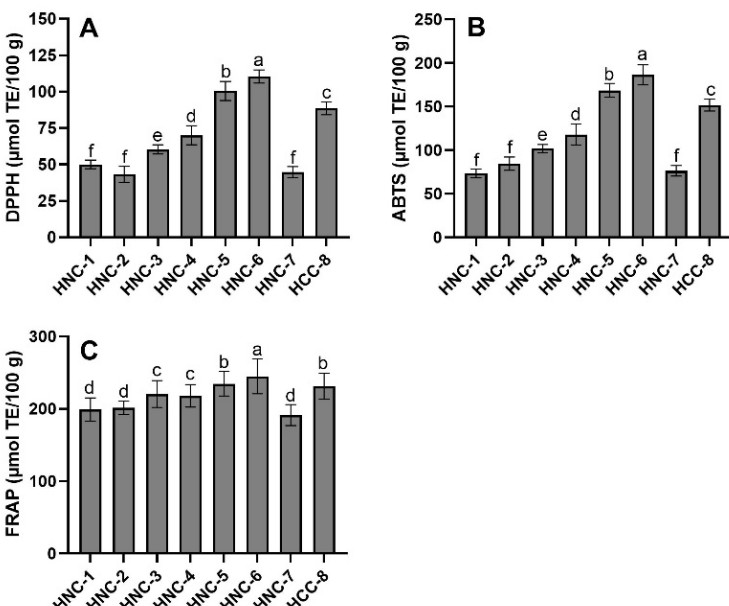

**Figure 6.** Antioxidant capacity of DPPH (**A**), ABTS (**B**), and FRAP (**C**) in fruits of eight genotypes of habanero pepper grown in the southeast of Coahuila. Values are the average of four replications. Means (*n* = 4). Bars represent the standard deviation of the mean. Different letters in each bar mean that the genotypes were statistically different (Tukey, $p \leq 0.05$).

Previous studies have documented that there is a strong correlation between total phenol content and antioxidant activity, which could be due to the high content of flavonoid-type compounds that contribute to the antioxidant activity in different species of chili peppers [68,69]. Therefore, the results of this research support the notion that the antioxidant activity is strongly influenced by the concentration of polyphenols in the evaluated habanero pepper fruits.

## 4. Conclusions

The fruit yield components were similar between the evaluated genotypes. However, the genotypes that showed better agronomic performance for plant height, stem diameter and internode length were HNC-2, HNC-3, HNC-6, and HNC-7. The HCC-8 genotype stood out in attributes related to the root system. In relation to the content of phytochemicals in the fruits, it was confirmed that there is a variability between genotypes. Regarding the content of capsaicinoids, total phenols, total flavonoids, and total anthocyanins and the antioxidant capacity, the HNC-5, HNC-6, and HCC-8 genotypes demonstrated the highest levels. Therefore, the genotypes that stood out in terms of agronomic performance (HNC-2, HNC-3, HNC-6, and HNC-7) and the content of bioactive compounds in the fruits (HNC-5, HNC-6, and HCC-8), will be used for the generation of inbred lines to obtain hybrids that can be cultivated and used in the agri-food industry in the southeastern region of Coahuila.

**Author Contributions:** Conceptualization, N.C.-M. and J.I.G.-L.; data curation, N.C.-M., A.F.-N. and J.I.G.-L.; funding acquisition, N.C.-M., N.R.-T., A.F.-N. and J.I.G.-L.; investigation, N.C.-M., P.Á.-V. and J.I.G.-L.; methodology, N.C.-M., X.R.-C., M.A.T.-T., G.N.-M. and P.R.-S.; project administration, J.I.G.-L. and N.C.-M.; resources, A.F.-N., N.R.-T., N.C.-M., V.V.-C. and J.I.G.-L.; supervision, N.C.-M., P.R.-S. and J.I.G.-L.; validation, N.C.-M., A.F.-N., P.Á.-V. and J.I.G.-L.; writing—original draft, N.C.-M. and J.I.G.-L.; writing—review and editing, J.I.G.-L. All authors have read and agreed to the published version of the manuscript.

**Funding:** This research was supported by the Research Direction of the Universidad Autónoma Agraria Antonio Narro (projects and grant number 38111-425105001-2446 and 38111-425105001-2166.

**Institutional Review Board Statement:** Not applicable.

**Informed Consent Statement:** Not applicable.

**Data Availability Statement:** All of the data are incorporated in the manuscript.

**Acknowledgments:** All authors wish to thank the Consejo Nacional de Ciencia y Tecnología (CONA-CYT) and the Support from the Research Direction of the Universidad Autónoma Agraria Antonio Narro, for the funding granted for the development of this research.

**Conflicts of Interest:** Authors have declared no conflict of interest.

**Sample Availability:** The seeds of the evaluated habanero pepper genotypes are available from the CCDTS.

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
