# Peer review of "Agronomic Performance, Capsaicinoids, Polyphenols and Antioxidant Capacity in Genotypes of Habanero Pepper Grown in the Southeast of Coahuila, Mexico"

_horticulturae, doi:10.3390/horticulturae7100372_

Round 1
Reviewer 1 Report
In the present manuscript authors describe the investigation of the “Agronomic performance, capsaicinoids, polyphenols and antioxidant capacity in genotypes of habanero pepper grown in the Southeast of Coahuila, Mexico”.
Quite a lot of experiments were involved in these studies; thus, the work is solid.
The introduction is clear and shows the literature data about the man subject. Methodology is adequate.
Results seem fine and appear correct.
The researchers used three methods to determine antioxidant activity, the results were consistent across the three trials, and consistent with the quantification of phenolic compounds. Which led me to consider these data quite solid. The other analyzes also led to satisfactory results.
However, the work does not bring innovations, the analyzes were incremental, used to help make decisions about how to select habanero pepper genotypes that can be used in breeding programs to increase the yield potential and the content of bioactive compounds in the fruits. Then, the work represents a modest contribution as innovations, but considering the aspect of applied results in agronomy, it can be accepted. I suggest accepting the work for publication as it was submitted.
Author Response
We thank the reviewer 1 for the time devoted to reviewing the document. We appreciate your valuable comments and your suggestion to accept the document as it was sent for publication in Horticulturae.

Reviewer 2 Report
- Although the abstract already has an introduction, problem statement, statement of possible solution selection, and final conclusion, no clear-cut future recommendations, and some quantitative data are provided to attract the interest of the reader. clarify these in 1-2 lines.
-
The bioactive compounds including capsaicinoids, polyphenols, and antioxidant capacity of habanero pepper need more highlighting in the introduction.
-
In general, this manuscript is easy to read and understand, it highlights the importance of the regional trend of habanero pepper, but it needs more about the importance of habanero pepper ingredients.
-
Plant height is missing in agronomic parameters in the material and methods section.
-
June-13 was the day of transplanting, September-10 was the day of the first harvest, what about sowing day? The authors mentioned that was 54 days before transplanting, but what is the sowing day exactly?
-
Table 1: Title should be placed above with one space.
-
Table 2: Some numbers with one digit and some with two and others without. Make numbers with one style.
-
Conclusion and Future Directions: It can be strengthened by citing specific lacunae and pathways to meet that.
-
References: Check DOI of all references because it is missing in many of them, and references should follow journal style.

Author Response
We thank the reviewer 2 for the valuable time spent reading and making constructive comments on this manuscript. We have followed all comments and the corresponding responses are displayed after each comment. We hope that the reviewer will find the revised version of our manuscript suitable for publication in Horticulturae.
Although the abstract already has an introduction, problem statement, statement of possible solution selection, and final conclusion, no clear-cut future recommendations, and some quantitative data are provided to attract the interest of the reader. clarify these in 1-2 lines.
Reply: We appreciate reviewer 2 for this valuable comment, we agree with this suggestion. Lines 1 and 2 of the abstract now read:
The genetic improvement program of the Seed Technology Training and Development Center, works on the agronomic characterization and the content of bioactive compounds in eight genotypes of habanero pepper. The objective is to select genotypes with good agronomic performance that allow the generation of inbred lines to obtain hybrids.
The bioactive compounds including capsaicinoids, polyphenols, and antioxidant capacity of habanero pepper need more highlighting in the introduction.
In general, this manuscript is easy to read and understand, it highlights the importance of the regional trend of habanero pepper, but it needs more about the importance of habanero pepper ingredients.
Reply: We thank reviewer 2 for these comments, to highlight the content of bioactive compounds in the fruits of the habanero pepper in the introduction section, the following paragraphs were added in the lines 62-74:
Several studies have documented that the main phenolic compounds found in pep-pers are vanillic, caffeic, ferulic, p-coumaric and p-hydroxybenzoic acids [7]. Ferulic acid has antiradical properties and vanillic acid is used primarily as a flavor enhancer [8]. Also, it is estimated that capsaicin and dihydrocapsaicin represent 80% of capsai-cinoids and their amounts are largely determined by the level of pungency [9].
Capsaicinoids have anti-inflammatory activity, in addition to promoting energy consumption and suppressing fat accumulation [10]. The antioxidant activities of capsai-cinoids and their lack of acrimony make them attractive for potential applications in food and pharmacology [11]. It should be noted that there is a positive correlation between an-tioxidant activity and the concentration of phenolic compounds in habanero pepper fruits [4]. This suggests that programs aimed at the genetic improvement of horticultural crops should consider the development of peppers with a high concentration of bioactive com-pounds that can be used in the food and pharmacological industry.
Plant height is missing in agronomic parameters in the material and methods section.
Reply: We thank reviewer 2 for the correction. The determination of plant height has been included in the materials and methods section. It now reads: Plant height (cm) and root length (cm) were determined with a Truper tape measure (PRO- 5MEC, Atlacomulco, Mexico).
June-13 was the day of transplanting, September-10 was the day of the first harvest, what about sowing day? The authors mentioned that was 54 days before transplanting, but what is the sowing day exactly?
Reply: We thank reviewer 2 for these comments, in the Crop management section, the following sentence was included: The sowing of the seeds was carried out on April 20, 2020.
Table 1: Title should be placed above with one space.
Reply: We thank reviewer 2 for this observation, the change was made.
Table 2: Some numbers with one digit and some with two and others without. Make numbers with one style.
Reply: We appreciate reviewer 2 for this valuable comment, we agree with this suggestion. The table was corrected with only one style of number.
Conclusion and Future Directions: It can be strengthened by citing specific lacunae and pathways to meet that.
Reply: We thank reviewer 2 for these comments. To strengthen and better understand future directions, the following section was included at the end of the conclusions section: Therefore, the genotypes that stood out in terms of agronomic performance (HNC-2, HNC-3, HNC-6, and HNC-7) and the content of bioactive compounds in the fruits (HNC-5, HNC-6, and HCC -8), will be used for the generation of inbred lines to obtain hybrids that can be cultivated and used in the agri-food industry in the southeastern region of Coahuila.
References: Check DOI of all references because it is missing in many of them, and references should follow journal style.
Reply: We thank reviewer 2 for this observation. All references have been verified and are in accordance with the journal format. However, the references that do not have DOI, is why they do not have one assigned.
We thank reviewer 2 for the valuable time spent reading and making constructive comments on this document. We have followed all the comments, and the corresponding responses are shown after each comment. We hope the reviewer will find the revised version of our manuscript appropriate for publication in Horticulturae.

Reviewer 3 Report
Methods
- Lines 147-148: change “…pathological damages were selected for each of the genotypes [15], subsequently, they were washed…” to “…pathological damages were selected for each of the genotypes [15]. They were subsequently washed…”
- Line 154: fruit bark = exocarp? If so, I recommend rephrasing.
- Lines 164-167: “Subsequently, they were placed in a polyethylene bag to make a homogeneous mixture, at the end, samples were stored in an ultrafreezer (3003 Ultrafreezer Thermo Scientific, Waltham, MA, USA) at -80 °C until their analysis.” Is a run-on sentence and should be changed to “Subsequently, they were placed in a polyethylene bag to make a homogeneous mixture. Samples were then stored in an ultrafreezer (3003 Ultrafreezer Thermo Scientific, Waltham, MA, USA) at -80 °C until their analysis.”
- Line 194: “..based on what was established by…” changed to “… per…”
Figures / Tables
- Table 1: correct “Origen” column header to “Origin”
Author Response
We thank the reviewer 3 for the valuable time spent reading and making constructive comments on this manuscript.
Comments.
Lines 147-148: change “…pathological damages were selected for each of the genotypes [15], subsequently, they were washed…” to “…pathological damages were selected for each of the genotypes [15]. They were subsequently washed…”
Reply: We thank reviewer 3 for his suggestions and constructive comments. The change was made, and the paragraph now reads: From the last harvest, four replications of 50 uniform fruits without physical and pathological damages were selected for each of the genotypes [20]. They were subsequently washed with a 3% sodium hypochlorite solution and dried in a cool and ventilated place [21].
Line 154: fruit bark = exocarp? If so, I recommend rephrasing.
Reply: We thank reviewer 3 for these constructive suggestions to improve the content of this article. The change was made and now it reads: The chromatic evaluations were carried out on the exocarp of the fruit, with a Konica Mi-nolta colorimeter (CR-10, Tokyo, Japan), according to García-López et al. [4].
Lines 164-167: “Subsequently, they were placed in a polyethylene bag to make a homogeneous mixture, at the end, samples were stored in an ultrafreezer (3003 Ultrafreezer Thermo Scientific, Waltham, MA, USA) at -80 °C until their analysis.” Is a run-on sentence and should be changed to “Subsequently, they were placed in a polyethylene bag to make a homogeneous mixture. Samples were then stored in an ultrafreezer (3003 Ultrafreezer Thermo Scientific, Waltham, MA, USA) at -80 °C until their analysis.”
Reply: We thank reviewer 3 for these constructive suggestions to improve the content of this article. The change was made and the paragraph now reads: Subsequently, they were placed in a polyethylene bag to make a homogeneous mixture. Samples were then stored in an ultrafreezer (3003 Ultrafreezer Thermo Scientific, Wal-tham, MA, USA) at -80 °C until their analysis.
Line 194: “..based on what was established by…” changed to “… per…”
We thank reviewer 3 for the correction. The change was made and the statement now reads: The capsaicionoid content (mg kg–1) was transformed into Scoville Heat Units (SHU), based on what was established per Todd et al. [27].
Table 1: correct “Origen” column header to “Origin”
Reply: We thank reviewer 3 for pointing out this correction. The change was made.
We thank reviewer 3 for the valuable time spent reading and making constructive comments on this document. We have followed all the comments, and the corresponding responses are shown after each comment. We hope the reviewer will find the revised version of our manuscript appropriate for publication in Horticulturae.
